# A Comparative Study of Various Land Use and Land Cover Change Models to Predict Ecosystem Service Value

**DOI:** 10.3390/ijerph192416484

**Published:** 2022-12-08

**Authors:** Chaoxu Luan, Renzhi Liu

**Affiliations:** State Key Laboratory of Water Environment Simulation, School of Environment, Beijing Normal University, No. 19, Xinjiekouwai Street, Haidian District, Beijing 100875, China

**Keywords:** ecosystem service value, LULC change analysis, different cellular automata (CA)-based models, policy impact, Tongliao city

## Abstract

Ecosystem services are closely related to human well-being and are vulnerable to high-intensity human land-use activities. Understanding the evolution of land use and land cover (LULC) changes and quantifying ecosystem service value (ESV) are significant for sustainable development. In this study, we used land use and land cover data and other data from 2000 to 2020 to analyze the evolution of land use and land cover and ESV in Tongliao, China. With the goal of exploring the characteristics of different cellular automata (CA)-based models, CA-Markov, Future Land Use Simulation (FLUS), and Patch-generating Land Use Simulation (PLUS) models were used to simulate future land use and land cover, and the results were verified and compared. Considering the impacts of policies for capital farmland (CF) and ecological protection red line (EPRL) in the context of territorial spatial planning, four scenarios (inertial development, S1; CF, S2; EPRL, S3; EPRL and CF, S4) were set. The results showed that from 2000 to 2020, farmland and built-up land increased the most (341.18 km^2^ and 220.56 km^2^), while grassland had the largest decrease (380.08 km^2^). The main mutual transitions were from grassland and farmland. The total ESV showed a decreasing trend (from 52,364.56 million yuan to 51,620.62 million yuan). The simulation results for 2035 under four scenarios were similar, where farmland would decrease the most (96.81 km^2^). The ESV in 2035 would decrease from 51,620.62 million yuan to 51,541.12 million. In addition, under scenarios for the impact of policy, the land showed a trend of scattered expansion. This study provides a scientific basis for making regional sustainable development policy decisions and implementing ecological environmental protection measures.

## 1. Introduction

Ecosystem services are the services and products that humans obtain directly or indirectly from the structure, process and function of the ecosystem, including four major functions: supply, regulation, support and cultural services [1,2]. Natural ecosystems provide many ecosystem services to humans [3]. Ecosystem services are closely related to human health by protecting human productivity and quality of life and are invaluable while protecting human well-being [4,5]. In 1997, Costanza pointed out the importance of the assessment of ecosystem services value (ESV) and established the first global ESV coefficient, which calculates the change in ESV in terms of monetary units [6]. After that, this method was widely used in the formulation of ecological restoration and sustainable development policies [7,8]. Land use and land cover (LULC) influence the structure and function of ecosystems through biogeochemical cycles. It is the closest link between people and nature and is closely related to the achievement of sustainable development goals [9]. Humans can alter the structure and patterns of LULC through high-intensity land use activities, resulting in rapid urban expansion, heightened tensions between humans and the environment, and impacts on natural ecosystems [10,11,12]. Therefore, LULC is the main driving force of ecosystem services and plays a decisive role in maintaining ecosystem service functions. At present, a large number of studies have used LULC to estimate changes in ESV [13,14,15,16]. The simulation and prediction of LULC and ESV from the spatial level are of great significance for regional environmental planning and management and sustainable development.

The change in LULC is a dynamic process. For example, land use patterns will evolve in time and space in the process of urbanization and crop growth [17]. The LULC model quantifies the combined impact of environmental and human drivers in the spatial pattern of LULC through information such as shape, fringe, and characteristics [18]. Therefore, the LULC change model is an effective and repeatable tool that plays an important role in environmental impact, land use planning and sustainable development [19,20]. Many methods have been used to study the simulation and prediction of LULC. Traditionally, Cellular Automata (CA) have been widely used for LULC modeling. CA is a spatially and temporally discrete model that evolves on a grid of discrete time steps according to the dynamic transition rules of neighboring cells [21]. The probability that a cell will change is calculated by transformation rules calculated from expertise [22]. In conclusion, CA is powerful in LULC transitions and is, therefore, widely used to model various real-world systems.

CA-based LULC change models containing fixed conversion rules mainly include the SLEUTH model [22] and the CLUE model [23]. These transition rules are a priori, which will limit the generality of regions that follow other rules. On this basis, a large number of studies have improved the CA-based LULC change model. For example, the combination of the CA model with artificial intelligence algorithms (including decision trees [24], support vector machines (SVM) [25], artificial neural networks (ANN) [26], and random forest (RF) [27]) increases the flexibility and accuracy of the model. However, few scholars use multiple CA-based LULC change models to analyze their strengths and weaknesses from multiple perspectives in one study area. Meanwhile, few existing studies have considered the impact of territorial spatial planning on the temporal and spatial evolution of ESV based on LULC, which is beneficial to sustainable development and ecological environment management in a study area.

By considering all of these points, in this study, CA-Markov, FLUS, and PLUS models were used to simulate LULC, and the results were verified and compared. In recent years, these three models have been widely used and are representative; it is necessary to carefully analyze their similarities and differences, advantages and disadvantages within a study area. This study takes Tongliao, which is located in the hinterland of Horqin Sandy Land, as the case area. There are famous grasslands, forests, deserts and ecological protection and restoration projects. It is of strategic significance to study the spatiotemporal evolution of LULC and ESV in Tongliao. The aims of this study were to: (1) analyze the historical LULC transition and ESV changes in Tongliao from 2000 to 2020; (2) compare and validate simulation results of different methods; (3) predict the LULC and ESV of Tongliao at a grid scale in 2035 considering potential driving factors and impact of territorial spatial planning. In order to achieve sustainable development goals, to ensure food security and ecological functions, this study considered the capital farmland (CF) and ecological protection red line (EPRL) under the policy impact of territorial spatial planning [28].

## 2. Materials and Methods

The flowchart of the methodology of this study is shown in Figure 1. The LULC dataset, driving factors data, and ESV coefficients data provided the basis for analysis. In the first part of the research process, we selected the LULC data from 2000, 2005, 2010, 2015, and 2020 to analyze the historical evolution characteristics of LULC. Besides, in the process of LULC-based ESV evaluation, we revised the ESV coefficient of Tongliao. In the second part, we simulated LULC in 2020 using the CA-Markov, FLUS and PLUS models, respectively. Then, we compared and verified the simulation results of the three models according to the real LULC situation of Tongliao in 2020 and selected the most advantageous model. Finally, in the third part, the model chosen in the second part was used for future LULC and we analyzed the evolution characteristics of LULC and ESV under multiple scenarios in 2035. Four simulation scenarios were set up in this study according to the policy impact of the CF and EPRL.

### 2.1. Study Area

Tongliao (42°15′~45°59′ N, 119°14′~123°43′ E) is located in the eastern part of Inner Mongolia, which is the hinterland of Horqin Sandy Land (Figure 2). The terrain in the south and north of Tongliao is high, while the terrain in the middle is low and flat, and the whole city is saddle-shaped. Tongliao has a total area of 58,800 square kilometers. There are famous grasslands, forests and deserts in the city, with various land types and broad development space. Tongliao has a superior geographical location and is an important strategic node for China to implement the “the belt and road initiative” and for Inner Mongolia to promote its opening to the north. In addition, Tongliao is not only a major agricultural area but also an important animal husbandry area, with obvious characteristics of interlaced agriculture and animal husbandry. In recent years, due to high agricultural income, farmers reclaim land without permission, which leads to a decrease in grassland area year by year and a continuous increase in cultivated land area, which also aggravates the trend of water shortage in the city. The decline in groundwater level leads to the further aggravation of soil desertification and a fragile ecological environment, so it is urgent to change the form of land use.

### 2.2. Data Sources

The LULC, gross domestic product (GDP), population, annual precipitation, and annual mean temperature data were obtained from the Resources and Environment Data Center of the Chinese Academy of Sciences (http://www.resdc.cn (accessed on 30 July 2022)). The LULC data were classified into farmland, grassland, woodland, water, built-up land, and unused land. Groundwater depth data were derived from groundwater monitoring data of the local water conservancy department in Tongliao and were obtained by interpolation processing. The terrain elevation, slope, and topographic relief data were derived from a 90 m × 90 m digital elevation model of Tongliao using a surface analysis process. They were collected from the geospatial data cloud (http://www.gscloud.cn (accessed on 30 July 2022)). Soil data were derived from the National Earth System Data Center (https://soil.geodata.cn/ztsj.html (accessed on 30 July 2022)). Night-time light data were derived from Chen et al. [29]. The road data were obtained from OpenStreetMap (https://www.openstreetmap.org (accessed on 30 July 2022)). The ecological protection red line and capital farmland protection area were digitized from maps based on the territory spatial planning for Tongliao. In order to ensure the consistency of data analysis, all data were resampled to 100 m × 100 m spatial resolution with a range of 3615 × 3763 grid cells.

### 2.3. LULC Simulation Model

#### 2.3.1. CA-Markov Model

In this study, we used a CA filter and a Markov chain model in IDRISI 17.0 software to predict LULC. The CA-Markov model can be applied and evaluated on the data of the previous period and predict the pattern of LULC in the next period through Geographic Information Systems [30]. The prediction of LULC using IDRISI software mainly consists of three steps: (1) we convert the data format in IDRISI software, convert ASCII files to raster files, and reclassify the converted files. (2) According to the reclassified land use data of the previous period and the latter period, a Markov matrix is obtained. The obtained Markov matrix records the probability of transferring from each land use type to other land use types in the next period, and the area transfer matrix can be obtained. (3) The file generated in the previous step based on the land use data of the two previous periods is used as the transition suitability map. The future LULC is then simulated based on the land use data, the transition probability matrix, the number of cellular automata cycles, and the set results of the neighborhood. For the setting of the neighborhood structure in the CA model, the default in the software is 5 pixels × 5 pixels.

In this study, a total of two simulations of LULC were performed using this method. For the first simulation, we used the land use data in 2010 as the base map and used the land use data in 2010 and 2015 to calculate the Markov transition matrix. Then, we combined the transition probability matrix and related data into the spatial operator of the IDRISI cellular automata, and on this basis, simulated the LULC in 2020, and compared the LULC simulation results in 2020 with the real observation data in 2020, to verify the accuracy of the CA-Markov model in IDRISI 17.0 software. For the second simulation, we used the land use data in 2015 as the base map and used the land use data in 2015 and 2020 to calculate the Markov transition matrix. Finally, the LULC in 2035 for the long-term forecast was based on this.

#### 2.3.2. FLUS Model

The FLUS model is established under the CA framework and has been widely used in LULC change modeling. The framework sets that the overall growth probabilities of each land use type (OP) are a product of the LULC change potential (P); also termed the probability-of occurrence), neighborhood effect (Ω), adjustment factor (inertia) and development restriction (R). The formula for calculating the overall growth probabilities of each land use type ${OP}_{i,k}^{d=1,t}$ of land use type *k* is as follows:(1)$${OP}_{i,k}^{d=1,t}{=P}_{i,k}^{d=1}\times Inertia_{k}^{t}{\times \Omega }_{i,k}^{t}\times R,$$
where $P_{i,k}^{d=1}$ is the probability-of-occurrence of land use type *k* in cell *i*; $Inertia_{k}^{t}$ is an adaptive driving coefficient, which represents the future impact on the demand for the same land use type and depends on the gap between the amount of land use type *k* and the target demand at iteration *t*; $\Omega_{i,k}^{t}$ is the neighborhood effect and R is the development restriction. Probability-of-occurrence *P* represents site conditions for LULC change according to a set of spatial driving factors. Specifically, for each macro area, grid samples are collected to train an artificial neural networks (ANN) classifier to generate city classification probabilities, with spatial driving factors as input features. The trained ANN classifier is then used to estimate the probability of occurrence for each grid in the entire region. Artificial Neural Networks (ANNs) are a family of machine learning models inspired by biological neural networks. Its strength lies in its ability to learn and fit complex relationships between input data and training targets, which have been successfully applied to the modeling of various nonlinear geographic problems [31,32]. Generally, an ANN consists of three types of layers: input layer, hidden layer and output layer. The formula for calculating the probability-of-occurrence of land use type *k* (*P* (*p*, *k*, *t*)) can be expressed as follows:(2)$$\begin{array}{l} P\left(p,k,t\right)=\underset{j}{\sum }w_{j,k}\times sigmoid\left(net_{j}\left(p,t\right)\right)\\ \;\;\;\;\;\;\;\;\;\;\;\;\;\;\;\;=\sum_{j}w_{j,k}\times \frac{1}{1+e^{-net_{j}\left(p,t\right)}}, \end{array}$$
where $w_{j,k}$ is the adaptive weight between the hidden and output layers, which is calibrated during training; $net_{j}\left(p,t\right)$ is the signal received by neuron *j* in the hidden layer and the sigmoid activation function effectively establishes the connection between the hidden layer and the output layer. So far, the ANN model has been established, which can be used to estimate the probability of occurrence of each land use type. The neighborhood effect considered in this study is similar to that of traditional CA models. $\Omega_{i,k}^{t}$ is the neighborhood effect of cell *i*, which is the coverage of land use type *k* in the following neighborhood:(3)$$\Omega_{i,k}^{t}=\frac{con\left(c_{i}^{t-1}=k\right)}{n\times n-1}\times w_{k},$$
where $con\left(c_{i}^{t-1}=k\right)$ represents the total number of grid cells occupied by land use type *k* at the last iteration of the *n* × *n* window, and *w* is the weight between different land use types. The default value of *w* is 1. The inertia coefficient is used to adjust the growth rates of the various land use types in the simulation and to facilitate convergence towards the expected quantity. If the development trend of a particular land use type contradicts the macro demand, the inertia coefficient will dynamically correct the land use trajectory in the next iteration. The calculation method of inertia coefficient $Inertia_{k}^{t}$ is as follows:(4)$$Inertia_{k}^{t}=\{\begin{array}{c} Inertia_{k}^{t-1}\;if\;\left|D_{k}^{t-1}\right|\leq \left|D_{k}^{t-2}\right|\\ Inertia_{k}^{t-1}\times \frac{D_{k}^{t-2}}{D_{k}^{t-1}}\;if\;0>D_{k}^{t-2}>D_{k}^{t-1}\\ Inertia_{k}^{t-1}\times \frac{D_{k}^{t-1}}{D_{k}^{t-2}}\;if\;D_{k}^{t-1}>D_{k}^{t-2}>0 \end{array},$$
where $D_{k}^{t-1}$ and $D_{k}^{t-2}$ are the difference between the current land use type *k* and the future demand for the same land use type at iterations t−1 and t−2. In addition, we consider the policy impact in the development restriction (R) module of the FLUS model. This study focuses on the spatialization of territorial spatial planning to better reflect the impact of policies on future LULC and ecosystem services value. The development restriction (R) factor has binary values and the restrictive effect of the policy is considered. The value with 0 refers to the condition of no land use transition allowed (otherwise, 1).

#### 2.3.3. PLUS Model

The PLUS model is similar to the FLUS model, with CA as the framework, including neighborhood effect, adjustment factor and development restriction. This model is designed to improve the accuracy of LULC simulations by more accurately simulating the nonlinear relationships inherent in LULC patch-level variation. The PLUS model integrates a rule mining framework based on land expansion analysis strategy (LEAS) and CA based on multi-type random patch seeds (CARS) [33]. On the one hand, LEAS simplifies the analysis of LULC change by obtaining patches for each changed land use type through source land use type data and transferred land use type data. This module mines the driving factors through the random forest algorithm and can obtain the probability of each type of LULC expansion. LEAS can better demonstrate the spatiotemporal evolution of LULC. On the other hand, the CARS module has a “bottom-up” constraint, retaining the roulette competition mechanism and adaptive inertia mechanism of the FLUS model, while adding a multi-type random patch seeding mechanism based on the decreasing threshold rule. This decreasing threshold rule makes cells with a higher overall probability most likely to change first. When a land use type wins a round of competition, the candidate land use type *c* selected by the roulette wheel is evaluated according to a decreasing threshold τ with the following formula:(5)$$If\;\sum_{k=1}^{N}\left|G_{c}^{t-1}\right|-\sum_{k=1}^{N}\left|G_{c}^{t}\right|<Step\;Then,l=l+1,$$
(6)$$\{\begin{array}{c} Change\;P_{i,c}^{d=1}>\tau \;and\;TM_{k,c}=1\\ No\;change\;P_{i,c}^{d=1}\leq \tau \;or\;TM_{k,c}=0 \end{array}\;\tau =\delta^{l}\times r1,$$
where *Step* is the step size of the PLUS model; *δ* is the decay factor of the decreasing threshold *r*1; *r*1 is a normally distributed random value; *l* is the number of decay steps. $TM_{k,c}$ is the transition matrix that defines whether to allow the transition of land use type *k* to land use type *c*.

#### 2.3.4. Selection of Driving Factors

In the LULC model, the driving factors play a crucial role and directly affect the accuracy of the simulation. The factors that lead to the changes in LULC in the study area are complex and diverse and are the result of the combined effects of natural conditions, socioeconomic conditions, and human factors. Natural conditions are the basic factors that determine the distribution of LULC, which are directly related to land availability. In addition, socioeconomic conditions and human factors also play an important role in the evolution of LULC. Considering the principles of data availability, spatial difference and comprehensiveness, 11 driving factors were selected to construct an indicator system in this study. These driving factors included population (POP), GDP, Night-time light (NL), Terrain elevation (TE), slope(SL), terrain relief (TR), soil organic matter (SOM), groundwater depth (GD), annual precipitation (AP), annual mean temperature (AMT), and proximity to road (PR).

#### 2.3.5. Verification of Different Models

In this study, we simulated the LULC of Tongliao in 2010 and compared the simulated LULC with actual LULC data in 2010 to verify the reliability of the simulation results. The Kappa index was used to test the accuracy by comparing the actual data and the simulation results with the following formula:(7)$$Kappa=\left(P_{0}-P_{c}\right)/(P_{p}-P_{c}),$$
where $P_{0}$ is the overall classification accuracy; $P_{c}$ is the actual simulation accuracy; $P_{p}$ is the ideal simulation accuracy. In general, the value of *Kappa* greater than 0.75 indicate high agreement between the actual and simulated degree, and the range of values from 0.4 to 0.75 shows generally high agreement, and those below 0.4 indicate poor agreement.

#### 2.3.6. Spatial Interpolation

Groundwater depth is one of the driving factors of this study, and spatial interpolation provides an effective method for converting discrete groundwater monitoring sites into continuous spatial surfaces. In this study, with the support of ArcGIS software, the ordinary Kriging interpolation method is used to interpolate the groundwater depth of 186 monitoring sites in Tongliao, and the interpolation accuracy is verified by the cross-verification method. Figure A1 in Appendix A shows the maps of groundwater depth. The Mean Error and Root Mean Square Error of the cross-validation are −0.0059 and 0.5976, respectively. The results of spatial interpolation can be used to predict the future LULC.

### 2.4. Simulation Scenario Setting

In order to better compare the impact of different policies in territorial spatial planning on the future LULC and ecosystem services value, four scenarios were set in this study. Scenario 1 (S1) is an inertial development scenario that does not consider the policy impact of territorial spatial planning. It represents the development of land use according to historical trends. The Markov model is used to calculate the area of each land use type. Scenario 2 (S2) and Scenario 3 (S3) are “single-policy element” scenarios. On the basis of S1, S2 only increases the consideration of the ecological spatial constraints of EPRL. We add EPRL maps of Tongliao in the process of model simulation. Similarly, S3 is a “single-policy element” scenario that only considers the ecological spatial constraints of CF. Scenario 4 (S4) is a “dual-policy element” scenario that considers both the EPRL and CF protection.

### 2.5. ESV Evaluation

Quantitative evaluation of ESV is of great significance for maintaining regional ecological security and promoting the coordinated development of the regional economy and environment. In order to calculate China’s ESV reasonably, Xie et al. proposed to determine China’s ESV coefficient based on the function of farmland grain production [34]. The equivalent ESV coefficient of the farmland grain production function is 1, and the coefficients of other functions are equivalent values with 1 as the standard. According to this method, ecosystem services are divided into nine categories: food production, raw material, gas regulation, climate regulation, waste treatment, water conservation, soil fertility maintenance, biodiversity protection, and recreation and culture. However, this coefficient is only suitable for use at the national level; in actual research, the provincial or local coefficient should be revised according to the characteristic factors of the study area. In this study, Tongliao was taken as a case area. In order to take into account regional differences, the weighted average method was used to correct the provincial coefficient differences of ESV in Inner Mongolia [35]. Table 1 shows the calculation results of ESV per unit area for different LULC types in Tongliao. In addition, we set the coefficient of built-up land to 0, because the expansion of construction land leads to the loss of ESV [36]. The ESV of each LULC type in each grid cell was calculated as follows [37]:(8)$$ESV_{ij}=\sum A_{ij}\times V_{j},$$
(9)$$ESV_{i}=\sum_{j=1}^{n}ESV_{ij},$$
where $ESV_{ij}$ is the ESV of the *j*-th LULC type in the *i*-th grid cell; $A_{ij}$ is the area of the *j*-th LULC type in the *i*-th grid cell; $V_{j}$ is the ESV coefficient of the *j*-th LULC type.

## 3. Results

### 3.1. Historical Changes of LULC and ESV

#### 3.1.1. Spatiotemporal Evolution of LULC

Figure 3 shows the spatial distribution of LULC in Tongliao from 2000 to 2020. The LULC area results for 2000, 2005, 2010, 2015, and 2020 showed that grassland was consistently the most extensive, followed by farmland and unused land (Table 2). Other, forest and built-up land and water were far less extensive. In general, the area of each type of land in Tongliao has not changed much in the 20 years from 2000 to 2020. Among them, the area of farmland and built-up land increased the most, increasing by 341.18 km^2^ and 220.56 km^2^, respectively. The area of unused land increased the least, by a total of 2.49 km^2^. On the contrary, the area of grassland decreased the most, reaching 380.08 km^2^. The area of water and forest decreased by 123.61 km^2^ and 60.54 km^2^, respectively. In addition, during these 20 years, the area of built-up land has continued to increase, while the area of the forest continued to decrease. Farmland increased continuously from 2000 to 2015 but decreased from 2015 to 2020. On the contrary, grassland and water decreased continuously from 2000 to 2015 but increased from 2015 to 2020. The changing trend of unused land during 20 years fluctuated.

Figure 4 shows the area where the major LULC changes occurred between 2000 and 2020, and it also highlights the major pattern of changes over this period using a Sankey diagram. From 2000 to 2005, the main LULC changes came from grassland, farmland, and unused land. Mainly, 313.47 km^2^ of the grassland was transferred into arable land, while 212.87 km^2^ of the arable land was transferred into grassland. In addition, 244.90 km^2^ of the grassland was transferred to unused land, and 235.20 km^2^ of the unused land was converted to grassland. During the 10-year period from 2005 to 2015, the transition area between different LULCs was small. Differently, since 2015, there has been a larger transition between various types of LULC in Tongliao, which is more than in the past 15 years. Among them, the most important is the mutual transition between grassland and farmland. There was 2877.23 km^2^ of grassland transferred into farmland, and 2831.9 km^2^ of farmland transferred into grassland. In addition, a large amount of grassland was transferred to forest and unused land, with 1093.1 km^2^ and 1889.23 km^2^, respectively. There was also some farmland and grassland transferred into built-up land.

#### 3.1.2. Temporal Variations in ESV

Based on the revised value coefficients of various ESVs for Tongliao, we calculated the ESV in Tongliao. The changes in the value of provisioning services, regulating services, supporting services, and cultural services jointly affected the total ESV. Table 3 shows the changes in ESV in Tongliao from 2000 to 2020. The value of waste treatment, water conservation, biodiversity protection, and soil fertility maintenance was the largest, implying that regulating and supporting services played a dominant role. From 2000 to 2020, the ESV in Tongliao decreased continuously from 52,364.56 million yuan to 51,620.62 million yuan, a total of 743.94 million yuan. Among them, the value of water conservation and soil fertility maintenance decreased the most, by 237.09 and 161.75 million yuan, respectively. Only the value of food production increased. It increased by 15.65 million yuan from 2000 to 2005 and decreased continuously by 8.05 million yuan from 2005 to 2020. In addition, the values of gas regulation, climate regulation, biodiversity protection, and recreation and culture decreased continuously from 2000 to 2020. Differently, the values of water conservation and soil fertility maintenance decreased continuously from 2000 to 2015, while increasing from 2015 to 2020. Besides, the changing trend of the values of raw material and waste treatment was similar to the value of food production.

### 3.2. Comparison and Verification of Simulation Results of Different Methods

In this study, we used the CA-Markov model in IDRISI 17.0, FLUS model, and PLUS model to simulate the LULC of Tongliao in 2020, respectively. By importing the parameters required by three models and the Tongliao LULC data of 2015, we obtained the comparison maps of the real situation and LULC simulation results of three different models for 2020. We evaluated the spatial similarity between the simulated and actual LULC based on the overall accuracy (OA) and kappa coefficient. The value of the kappa coefficient is between 0 and 1. The larger the kappa coefficient is, the more accurate the simulation is. The distribution maps and the validated coefficients of LULC simulation in Tongliao using different models are shown in Figure 5. For the CA-Markov model in IDRISI 17.0, the final verification yielded an OA value of 0.7832 and a kappa coefficient of 0.6631. The kappa < 0.75, shows the agreement is general. In addition, for the FLUS model, the OA value is 0.8724 and the kappa coefficient is 0.7613, while the OA value and the kappa coefficient of the PLUS model are 0.8726 and 0.7622, respectively. In conclusion, the accuracy of the FLUS model and the PLUS model is similar, and both are higher than the CA-Markov model in IDRISI 17.0. Besides, the PLUS model is able to mine the driving factors of each type of LULC change individually, in this study, through the comparison of the three models, the PLUS model was finally chosen to predict the future LULC.

### 3.3. Driving Factors Analysis of LULC Evolution

In the PLUS model, the contributions of the driving factors to each land use type can be trained separately using the LEAS, making the driving factors analysis of LULC evolution more explicit. In this study, the driving factors of land expansion to all six land use types were analyzed. Figure 6 shows the contributions of the driving factors and the growth probabilities of all LULC types. In all six land use types, the areas of grassland with a high growth probability were widely distributed. The areas with high growth probability in farmland were also widely distributed. On the contrary, the areas with high growth probability in water and built-up land were few and scattered. Besides, the areas with high growth probability in the forest were mainly distributed in the northwest part of Tongliao, while the expansion of the unused land mainly occurred in the central and southern parts. In addition, the expansion of farmland, grassland, built-up land, and unused land was mainly influenced by population and GDP. The elevation of the terrain, terrain relief, and annual mean temperature played important roles in the expansion of the water and forest. In general, human activities and economic factors largely affect the land expansion of various types in Tongliao. Some natural factors such as the elevation of the terrain also played important roles.

### 3.4. LULC and ESV Predictions under Multiple Scenarios

In this study, the Markov model was used to predict the future land use demand in Tongliao. By using the land use demand calculated using the Markov model, the future LULC spatial distribution was simulated using the PLUS model constructed in this study. Table 4 shows the simulated results of LULC in 2035 compared to that in 2020. Figure 7 shows the LULC maps for 2035 under different scenarios. The LULC spatial simulation characteristics under S1, S2, S3, and S4 are generally similar. The area of each type of land expansion is extremely small. Therefore, in Figure 6, we can hardly see the spatial distribution of each type of land expansion. According to the simulation results, farmland will decrease the most in 2035, with a total decrease of 96.81 km^2^. Forests will decrease by 48.21 km^2^. In addition, the other four types of LULC will increase. Among them, the unused land will increase the most, which is 66.51 km^2^. Besides, in the predicted results of land expansion, unused land is also the type of LULC with the largest expansion area. As the total area decreases, the area of farmland and forest expansion is also the least. The increased area and expanded area of grassland are the same, indicating that in the simulation results, from 2020 to 2035, no grassland will be transferred to other types of land.

Based on the LULC simulation of Tongliao in 2035, we predicted the ESV of Tongliao in 2035 under different scenarios. Table 5 shows the predicted results of ESV in 2035 compared to that in 2020. Because the amount of each type of LULC in 2035 is the same for each scenario, the total ESV is the same for the four scenarios. According to the predicted results, the ESV in 2035 will decrease from 51,620.62 million yuan to 51,541.12 million yuan, a total of 79.5 million yuan. Among them, the value of waste treatment, gas regulation, and climate regulation will decrease the most, by 17.68, 16.27, and 15.2 million yuan, respectively. Only the value of soil fertility maintenance and water conservation will increase. The value of soil fertility maintenance will increase by 2.46 million yuan, while the value of water conservation will increase by 1.39 million yuan. In general, Tongliao’s ESV will not change significantly in 2035.

## 4. Discussion

So far, many countries have carried out research on LULC simulation. Unlike previous researchers, we simultaneously applied three different CA-based models to simulate LULC. At the same time, we focused on the impact of implementing territorial spatial planning on future LULC. The lower accuracy of the CA-Markov model in IDRISI may be due to the fact that the method mainly obtains the land use transition probability through the Markov matrix and generates a transition suitability map based on two-phase land use data. This method simplifies the generation of transition suitability maps without separate analyses of the driving factors of LULC changes. At the same time, since this method requires fewer types of data, the accuracy is greatly affected by the precision of land use data. On the contrary, the other two models, the FLUS model and the PLUS model, involve more complex algorithms. The calculation results of the Markov model are only used as the quantitative results of the land demand of the FLUS and PLUS models, and no spatialized layers are generated. In addition, the FLUS model and the PLUS model also have many similarities. Both of these two models are based on CA, and both include the neighborhood effect, adjustment factor and development restriction. We are also able to infer similar conclusions from the close accuracy of the two models. The biggest difference between the FLUS model and the PLUS model is the driving factors analysis module. In this module, the FLUS model contains the artificial neural networks classifier, while the PLUS model contains the random forest algorithm. Besides, the FLUS model mainly excavates the relationship between land use types and driving factors, while the PLUS model mainly excavates the relationship between land expansion and driving factors. In the PLUS model, the land expansion layer needs to be extracted based on the land use data of the two phases. Moreover, it can efficiently mine the driving factors of the LULC changes individually. Therefore, when choosing a model, we must consider not only the accuracy of the model but also whether there is a requirement to analyze the driving factors of each type of LULC change separately in the study.

Based on the results in Figure 7, we cannot clearly assess the impact of Tongliao’s territorial spatial planning on future land expansion. Because according to the forecast results, the amount of land expansion of each type in 2035 is extremely small without considering CF and EPRL. For example, in unused land with the largest amount of land expansion, the proportion of its expansion still does not exceed 1% of the total unused land (Table 4). In order to reflect the impact of CF and EPRL on future LULC simulations more intuitively, we also calculated the landscape metrics. Table 6 shows the overall landscape metrics in 2035 under different scenarios. At the landscape level, three landscape pattern indicators (the SHDI, SHEI, and CONTAG) were selected for analysis. The SHDI and SHEI were selected to study the landscape diversity, the CONTAG was selected to study the spatial distribution of each patch in the landscape. According to Table 6, the values of SHDI and SHEI are the same in these four scenarios. The order of the values of CONTAG from largest to smallest is S1 > S3 > S2 > S4. These subtle differences indicate that when CF and EPRL are considered, the original continuous expansion of land cannot expand within the restricted area, and the patches of land will be more fragmented.

Land-use change affects ESV by significantly altering the provision of ecosystem services. For example, urbanization-induced increases in large-scale built-up land can lead to a remarkable loss of ESV. This study found that the main change pattern of LULC in Tongliao from 2000 to 2020 was the mutual transfer between farmland, grassland and unused land. On the one hand, ecological civilization construction projects such as “returning farmland to forest and grassland” implemented in the region have led to the transition of farmland to grassland or forest, and the increase in forest, grassland, and water can effectively improve the regional ESV. On the other hand, the total amount of ESV in Tongliao has decreased slightly in the past 20 years, which shows that while the local ecological protection or restoration projects are implemented, there is also the phenomenon that forest and grassland are occupied by farmland and built-up land. The total amount of changes in each type of land and ESV is small, and the mutual transition of each type of land is large, which are the most significant characteristics of land use in Tongliao. In addition, the existing large amount of unused land in Tongliao shows that there is still great potential for the increase in ESV in this area, and there is still a lot of work to be conducted in the implementation of ecological protection and restoration projects.

In summary, in this study, under the context of ecological civilization and considering the elements of farmland-ecological land in Tongliao’s territorial spatial planning, we simulated the future LULC and predicted future ESV under multiple scenarios. Based on the results, we propose several suggestions for future land use in Tongliao. We should strictly abide by the ecological protection red line and capital farmland protection policies and protect ecological resources such as forests, grassland, and arable land. Besides, based on the simulation results, the future LULC under the impact of Tongliao’s territorial spatial planning will exhibit a trend of scattered expansion, which demonstrates the importance of innovative land use patterns, and the necessity to optimize land development activities in space and time. In addition, the rational use of unused land in the area and the implementation of ecological protection and restoration projects such as “returning farmland to forests and grasslands” are the keys to improving Tongliao’s ESV.

## 5. Conclusions

This study predicted the future ESV of Tongliao based on CA-based models and revised ESV coefficients. We analyzed the historical characteristics of LULC and ESV for the period 2000–2020. CA-Markov, FLUS, and PLUS models were used to simulate LULC and the results were verified and compared. Finally, based on the PLUS model, we successfully projected the spatial distribution of the LULC map and predicted the ESV for 2035 under multiple scenarios. The results are summarized as follows:
From 2000 to 2020, the main mutual transition came from grassland (increased by 341.18 km^2^) and farmland (decreased by 380.08 km^2^). The total ESV decreased continuously from 52,364.56 million yuan to 51,620.62 million yuan.The accuracy of the FLUS model (Kappa coefficient = 0.7613) and the PLUS model (Kappa coefficient = 0.7622) is similar, and both are higher than the CA-Markov model in IDRISI 17.0 (Kappa coefficient = 0.6631).Human activities and economic factors largely affect the land expansion of various types in Tongliao (population and GDP).In 2035, farmland will decrease the most (96.81 km^2^); the total ESV decreased from 51,620.62 million yuan to 51,541.12 million. Besides, the land showed a trend of scattered expansion under scenarios of policy impact.

The main contribution of our research is to carefully analyze the characteristics of different CA-based models and consider the policy impact of capital farmland and ecological protection red lines. Moreover, we provide effective suggestions for future land use and ESV improvement in Tongliao. The limitations of the study are also appreciated. The limitations in data acquisition prevented this study from considering many factors that would affect the simulation results. Besides, the limitations of the land demand predicted by the Markov model may be addressed in the next study.

## Figures and Tables

**Figure 1 ijerph-19-16484-f001:**
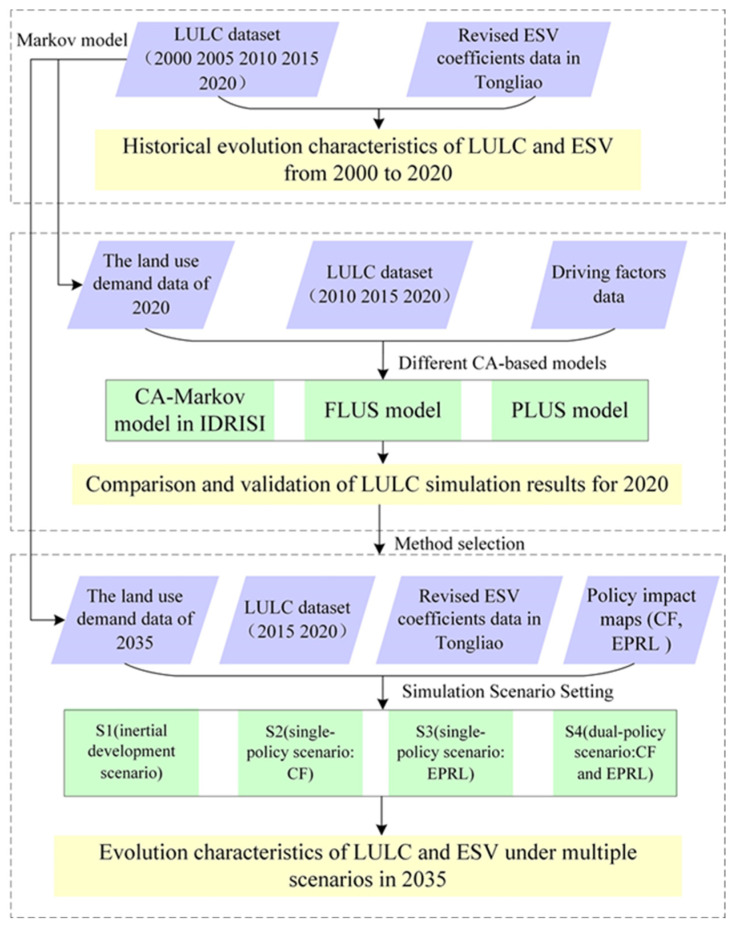
The flowchart for this methodology.

**Figure 2 ijerph-19-16484-f002:**
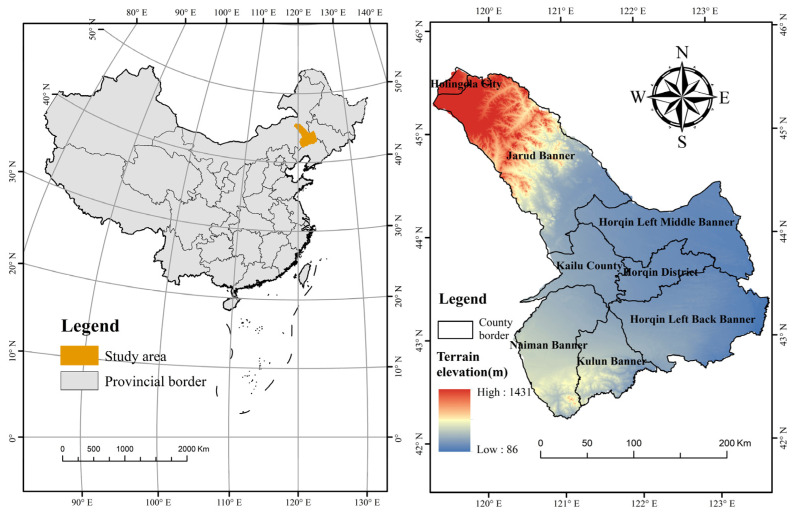
Location of the study area.

**Figure 3 ijerph-19-16484-f003:**
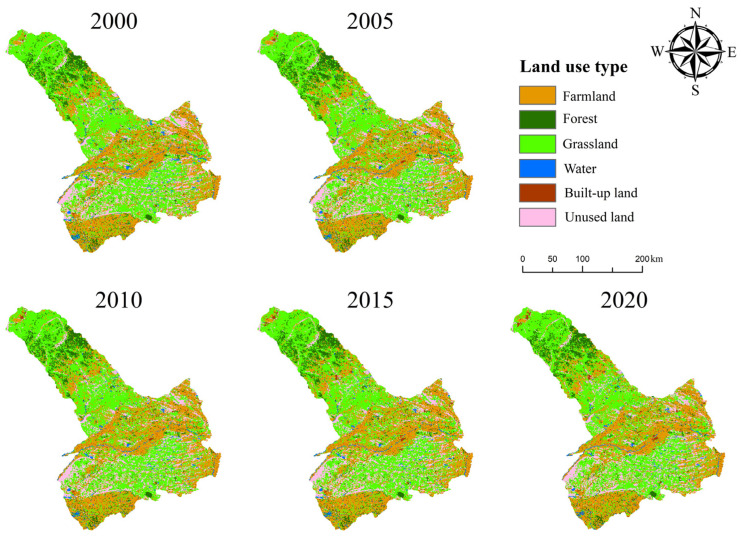
Spatial distribution of LULC from 2000 to 2020.

**Figure 4 ijerph-19-16484-f004:**
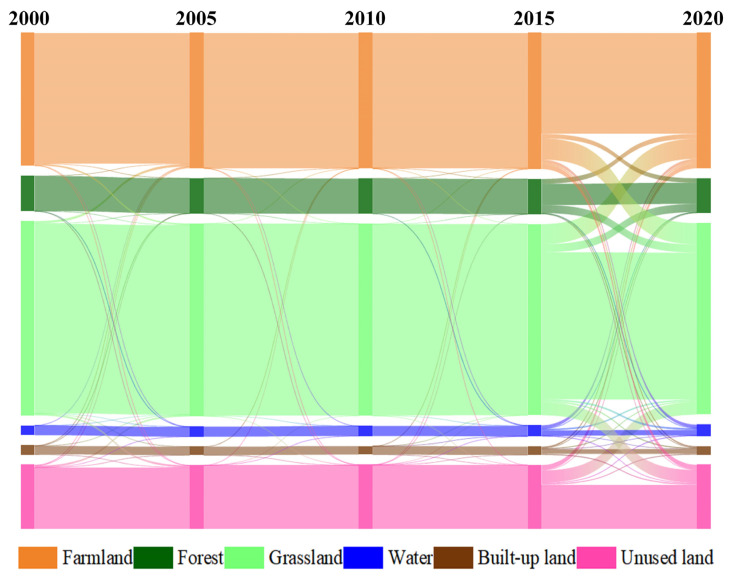
Land use transition from 2000 to 2020.

**Figure 5 ijerph-19-16484-f005:**
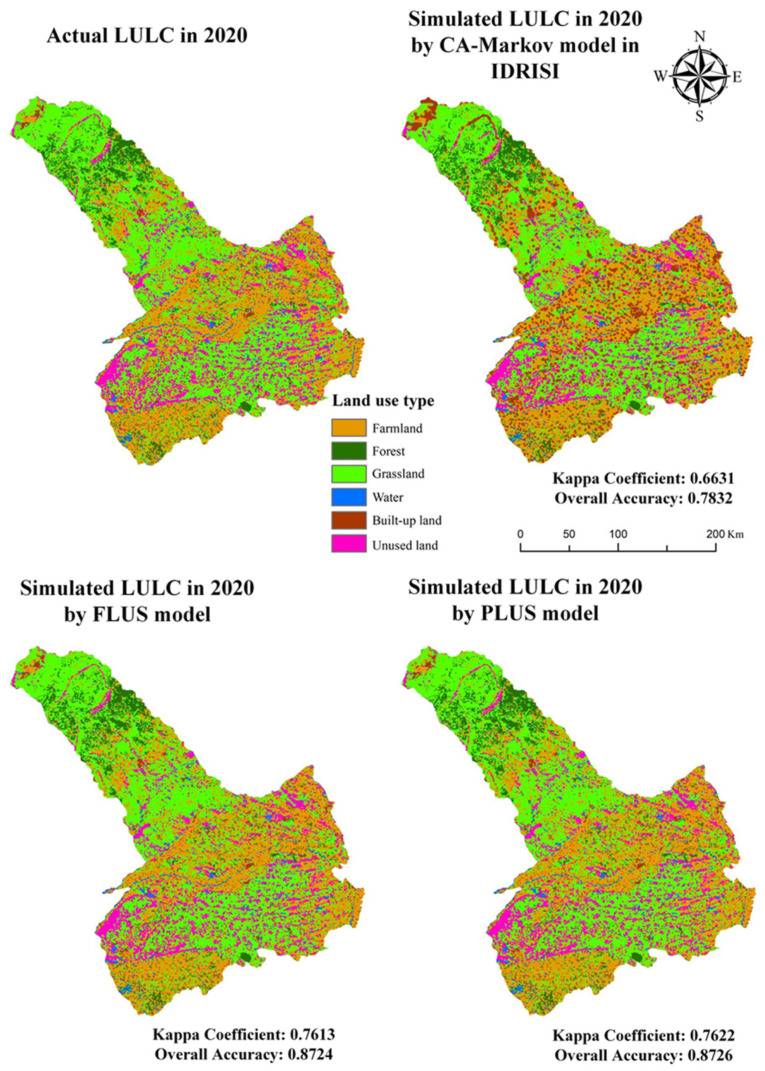
Distribution maps of LULC simulation in Tongliao using different models.

**Figure 6 ijerph-19-16484-f006:**
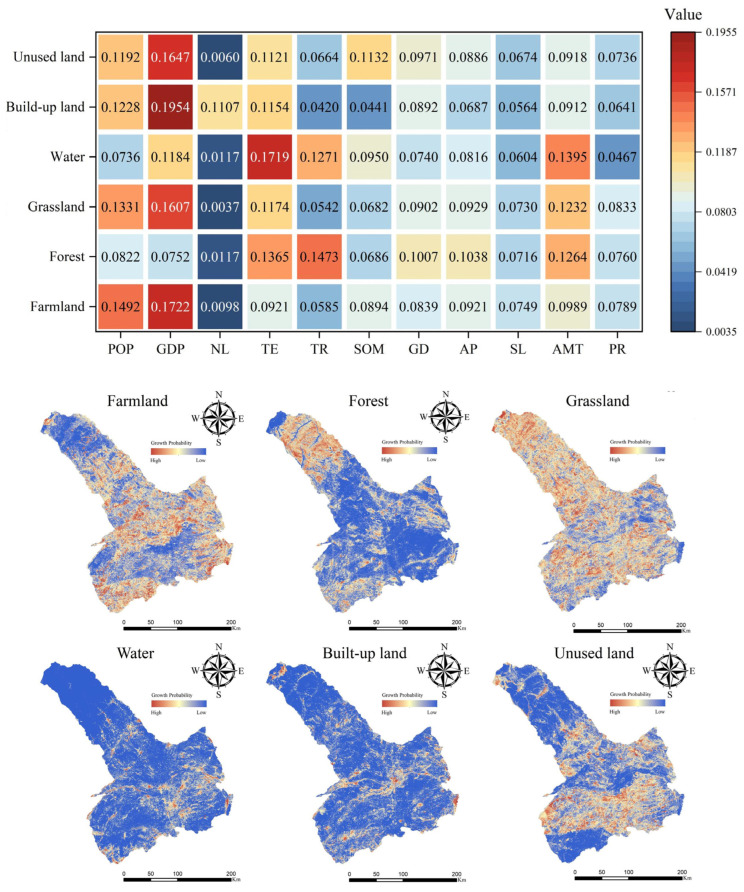
The contributions of the driving factors and the growth probabilities of all LULC types.

**Figure 7 ijerph-19-16484-f007:**
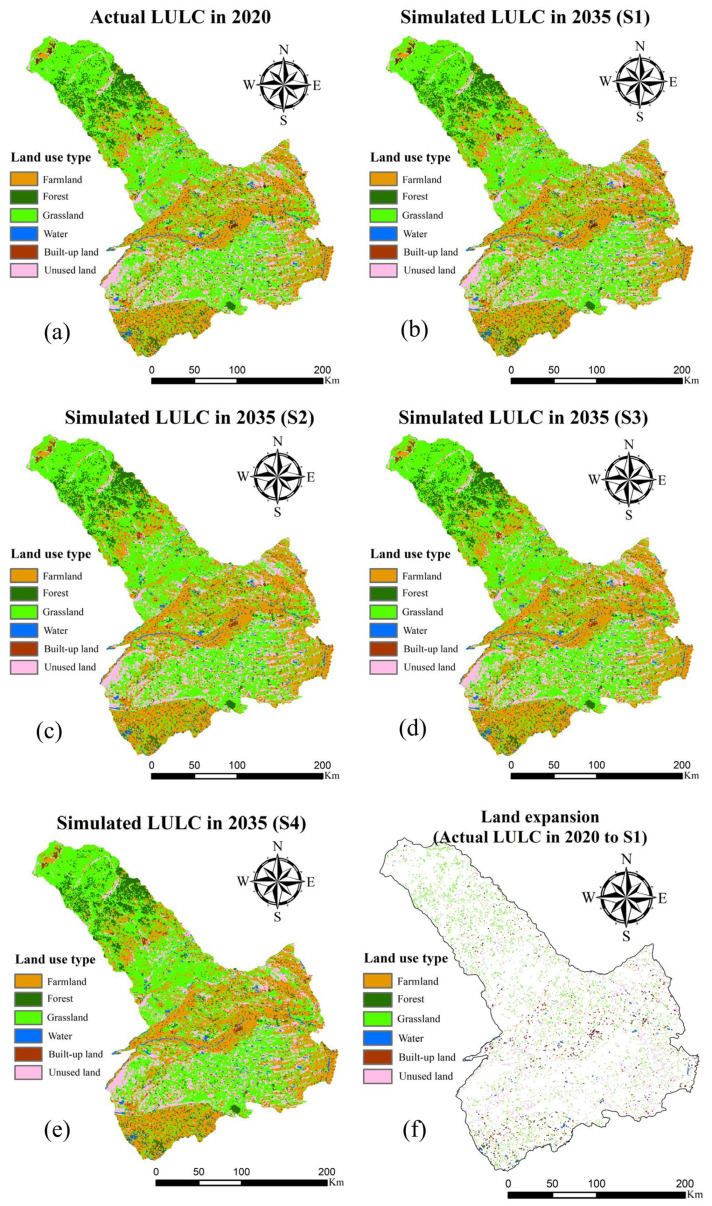
LULC maps for 2035 under different scenarios. (**a**) Actual LULC in 2020. (**b**) Simulation result of S1. (**c**) Simulation result of S2. (**d**) Simulation result of S3. (**e**) Simulation result of S4. (**f**) Land expansion of actual LULC in 2020 to S1.

**Table 1 ijerph-19-16484-t001:** ESV per unit area for different LULC types (RMB yuan·ha^−1^·yr^−1^).

Primary Classification	Secondary Classification	Farmland	Forest	Grassland	Water	Built-Up Land	Unused Land
Provisioning services	Food production	822.500	271.425	353.675	435.925	0	16.450
	Raw material	320.775	2451.050	296.100	287.875	0	32.900
Regulating services	Gas regulation	592.200	3553.200	1233.750	419.475	0	49.350
	Climate regulation	797.825	3347.575	1283.100	1694.350	0	106.925
	Waste treatment	1209.075	3306.450	1842.400	337.225	0	139.825
Supporting services	Water conservation	633.325	3364.025	1250.200	15,438.325	0	57.575
	Soil fertility maintenance	1143.275	1414.700	1085.700	12,214.125	0	213.850
	Biodiversity protection	838.950	3709.475	1538.075	2821.175	0	329.000
Cultural services	Recreation and culture	139.825	1710.800	715.575	3651.900	0	197.400

**Table 2 ijerph-19-16484-t002:** LULC change in the study area during different periods (km^2^).

LULC Type	2000	2005	2010	2015	2020	2000–2005	2005–2010	2010–2015
Farmland	17,508.93	17,834.52	17,877.61	17,932.10	17,850.11	325.59	43.09	54.49
Forest	4659.98	4654.55	4643.53	4641.51	4599.44	−5.43	−11.02	−2.02
Grassland	25,582.02	25,424.99	25,298.35	25,157.32	25,201.94	−157.03	−126.64	−141.03
Water	1256.17	1134.02	1132.01	1121.97	1132.56	−122.15	−2.01	−10.04
Built-up land	1304.49	1328.56	1344.28	1494.51	1525.05	24.07	15.72	150.23
Unused land	8476.59	8411.54	8492.40	8440.77	8479.08	−65.05	80.86	−51.63

**Table 3 ijerph-19-16484-t003:** Changes in ESV in the study area from 2000 to 2020 (×10^6^ yuan·yr^−1^)**.**

Primary Classification	Secondary Classification	ESV 2000	2005	2010	2015	2020	ESV Change 2000–2005	2005–2010	2010–2015	2015–2020	2000–2020
Provisioning services	Food production	2540.07	2555.72	2554.53	2553.44	2547.66	15.65	−1.19	−1.08	−5.78	7.59
	Raw material	2525.36	2526.09	2521.23	2517.85	2506.66	0.73	−4.86	−3.38	−11.19	−18.70
Regulating services	Gas regulation	5943.37	5935.90	5919.23	5903.66	5890.00	−7.47	−16.67	−15.57	−13.67	−53.37
	Climate regulation	6542.77	6525.39	6509.42	6492.74	6480.04	−17.38	−15.98	−16.68	−12.70	−62.73
	Waste treatment	8531.88	8535.49	8514.78	8493.66	8478.95	3.61	−20.70	−21.12	−14.71	−52.93
Supporting services	Water conservation	7862.90	7673.11	7653.66	7623.00	7625.80	−189.79	−19.45	−30.66	2.80	−237.09
	Soil fertility maintenance	7154.01	7022.83	7011.73	6988.99	6992.26	−131.18	−11.11	−22.73	3.27	−161.75
	Biodiversity protection	7765.49	7730.04	7712.18	7689.78	7678.41	−35.45	−17.86	−22.40	−11.37	−87.08
Cultural services	Recreation and culture	3498.70	3445.20	3435.71	3421.35	3420.83	−53.51	−9.48	−14.36	−0.53	−77.88
Total ESV		52,364.56	51,949.77	51,832.47	51,684.48	51,620.62	−414.79	−117.30	−147.99	−63.87	−743.94

**Table 4 ijerph-19-16484-t004:** Simulated results of LULC in 2035 compared to that in 2020.

LULC Type	2020	S1–S4	LULC Change	Land Expansion
Farmland	17,850.11	17,753.30	−96.81	10.89
Forest	4599.44	4551.23	−48.21	15.72
Grassland	25,201.94	25,248.91	46.97	46.97
Water	1132.56	1143.88	11.32	30.23
Built-up land	1525.05	1545.27	20.22	28.89
Unused land	8479.08	8545.59	66.51	71.85

**Table 5 ijerph-19-16484-t005:** Predicted results of ESV in 2035 compared to that in 2020 (×10^6^ yuan. yr^−1^).

Primary Classification	Secondary Classification	ESV 2020	S1–S4	ESV Change
Provisioning services	Food production	2547.66	2540.65	−7.01
	Raw material	2506.66	2493.68	−12.98
Regulating services	Gas regulation	5890.00	5873.73	−16.27
	Climate regulation	6480.04	6464.84	−15.2
	Waste treatment	8478.95	8461.27	−17.68
Supporting services	Water conservation	7625.80	7627.19	1.39
	Soil fertility maintenance	6992.26	6994.72	2.46
	Biodiversity protection	7678.41	7665.01	−13.4
Cultural services	Recreation and culture	3420.83	3420.03	−0.8
Total ESV		51,620.62	51,541.12	−79.5

**Table 6 ijerph-19-16484-t006:** Overall landscape metrics in 2035 under different scenarios.

Landscape Indicators Scenario	SHDI	SHEI	CONTAG
S1	1.3753	0.7676	54.3193
S2	1.3753	0.7676	54.0539
S3	1.3753	0.7676	54.1384
S4	1.3753	0.7676	54.0111

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
