# Peer review of "A Comparative Study of Various Land Use and Land Cover Change Models to Predict Ecosystem Service Value"

_ijerph, 2022, doi:10.3390/ijerph192416484_

Round 1

Reviewer 1 Report

First of all, I would like to say that this manuscript focuses on a very interesting research problem. The article covers the topics included in the main subjects and I recommend to be considered for the publication after a revision.

 TITLE

The article’s title is suitable with the content of the paper.

ABSTRACT

The abstract is well-designed and briefly express the present research thus being of interests and readable thus capturing the reader’s attention. It present in an appropriate manner the main research hypothesis, the problem statement, the methods and the main findings. However, you can't use an abbreviation before explaining it, so I suggest giving LULC in parentheses and inserting the full name into the text of the abstract.

KEY WORDS

The key words are appropriate to the present research and are clearly stated.

ORIGINALITY

The article meets a high level of originality argued by the main research theme and the research hypothesis. Furthermore, the originality of the paper is highlighted by the main results of the paper.

The authors construct a well-designed theoretical background closely related to the current specialised literature in the field. A short recommendation I would like to made, it is stated in the final part of this review form.

THE PAPER S STRUCTURE

The structure of the paper is correct in line with the journal standards and meet the publication requirements considering the paper logic. The objectives seem to be clear formulated as well as the investigation is drawn. The core argument of the paper illustrates the paper relevance and the research originality. The results are clearly express and well connected both to the theoretical framework and discussions.

THE METHODS

The methodological design is appropriate and the methods fit well to the present investigation. The methods used in the study are well expressed both in the graphical form as well as in the main text of the manuscript. Please refer to the NDVI Raster Function which can be used to detect and evaluate vegetation properties in a raster image. Is the LULC method a similar method?

Is the actual state of use on the map an important aspect in the LULC method and whether the records of planning purpose are taken into account because often the identification on the map does not coincide with the records of land and buildings (e.g. a meadow is visible on the map from the satellite image, but in the records it is a forest).

THE MAIN ANALYSIS

The main research is well design. The authors have carried out a lot of research, but the description mentions that data interpolation was carried out, although it is hard to find a stage, let alone its validation. It seems that the developed maps are rather raster visualizations derived from satellite images or photos from air raids. Please specify. Provide validation and interpolation methods used in the study.

Please refer to the rights restricted in the land and building register. Is it possible to apply layers with restrictions on use to the analyses mentioned in Ogryzek, M.; Klimach, A.; Niekurzak, D.; Pietkiewicz, M. Using Cartographic Documents to Provide Geoinformation on the Rights to Real Estate—Taking Poland as an Example. ISPRS Int. J. Geo-Inf. 2019, 8, 530. https://doi.org/10.3390/ijgi8120530...

CONCLUSIONS

The conclusions fit well summarising the main ideas of the present analysis. I support the idea of a scientific discussion, also critical of the research conducted.

THE GRAPHICAL SUPPORT

The graphical support is well formatted, appropriate illustrating the text content. However, in Figure 2, there is no need to add north to the map on the left because a grid is inserted. Pie charts are not clear and readable.

THE ENGLISH LANGUAGE

I think the English is ok as far as I could see. I enjoyed to read this paper in English and the language seems well but I think that an opinion of a native English speaker is welcomed. In other words, if the authors used a specialised proofreading services and they could prove this aspect I trust the opinion and the work of this proofread service. On the other hand, I put my trust regarding the English language on the journal editors but I repeat the language seems well.

RECOMMENDATIONS

Finally, I recommend the publication of this paper with some minor revision considering the above mentioned aspects. I want to see the revised version of this paper before publication for a final acceptance and to ensure that the revision has been completely and carefully made.

Reviewer 2 Report

1.Moderate English changes required

2.Refine the part of Introduction

Reviewer 3 Report

This paper uses a variety of CA models to predict land use change, and on this basis, takes Tongliao as the experimental area to evaluate ecological value. This paper has certain practical value, but the following problems need to be solved before publication.

Compare the Figure 2 and Figure 3, the distance of the scale bar was different. I think that one of the maps may be wrong.

Line 347-348, the presentation should be checked.

Figure 7, Is there a problem with the sixth picture, and should each sub-picture also be given a number?

The full text uses too many abbreviations. To improve the readability of the round text, it is suggested that a list of abbreviations can be added to the appendix.

The geographical meaning of Scenario 1-3 should be introduced in more detail, so it is recommended to refer to the newest REF.  https://doi.org/10.3390/rs14061452; https://doi.org/10.1016/j.scs.2022.104055; https://doi.org/10.3390/w14030402 . Although this work uses a different method to simulate the land use change, the writing way and the section organization can all be drawn on.

It is also recommended to revise the content of the review section. After all, if there is no literature in 2022, it is difficult to prove the novelty and innovation of this article, although the author may have done a lot of research.

It is advised to check the spelling of the full text in detail.
